# Real-World Use of Highly Sensitive Liquid Biopsy Monitoring in Metastatic Breast Cancer Patients Treated with Endocrine Agents after Exposure to Aromatase Inhibitors

**DOI:** 10.3390/ijms241411419

**Published:** 2023-07-13

**Authors:** Jesús Fuentes-Antrás, Ana Martínez-Rodríguez, Kissy Guevara-Hoyer, Igor López-Cade, Víctor Lorca, Alejandro Pascual, Alicia de Luna, Carmen Ramírez-Ruda, Jennifer Swindell, Paloma Flores, Ana Lluch, David W. Cescon, Pedro Pérez-Segura, Alberto Ocaña, Frederick Jones, Fernando Moreno, Vanesa García-Barberán, José Ángel García-Sáenz

**Affiliations:** 1Department of Medical Oncology, Hospital Clínico San Carlos, Instituto de Investigación Sanitaria Hospital Clínico San Carlos (IdISSC), 28040 Madrid, Spain; jesus.fuentesantras@uhn.ca (J.F.-A.); alicia.delunaaguilar@gmail.com (A.d.L.); ca.ramirezruda@gmail.com (C.R.-R.); paloma.flores@salud.madrid.org (P.F.); pedro.perez@salud.madrid.org (P.P.-S.); albertocana@yahoo.es (A.O.); fmorenoa@salud.madrid.org (F.M.); 2Experimental Therapeutics Unit, Hospital Clínico San Carlos, IDISSC and CIBERONC, 28040 Madrid, Spain; ilopez.7@alumni.unav.es; 3Molecular Oncology Laboratory, IdISSC, 28040 Madrid, Spain; anamr.hcsc@gmail.com (A.M.-R.); victor.lorca@salud.madrid.org (V.L.); 4Department of Clinical Immunology, Hospital Clínico San Carlos, IdISSC, 28040 Madrid, Spain; kissy.guevara@salud.madrid.org; 5Cancer Immunomonitoring and Immune-Mediated Diseases Unit, Hospital Clínico San Carlos, IdISSC, 28040 Madrid, Spain; 6Department of Pathology, Hospital Clínico San Carlos, 28040 Madrid, Spain; alejandro.pascual@salud.madrid.org; 7Medical Affairs Division, Sysmex Inostics, Inc., Baltimore, MD 21205, USA; preston.jennifer@sysmex-inostics.com (J.S.); jones.fred@sysmex-inostics.com (F.J.); 8INCLIVA Research Institute, Hospital Clínico Universitario de Valencia, 46010 Valencia, Spain; ana.lluch@uv.es; 9Division of Medical Oncology and Hematology, Princess Margaret Cancer Centre, University Health Network, Toronto, ON M5S A18, Canada; dave.cescon@uhn.ca

**Keywords:** breast cancer, liquid biopsy, ctDNA, endocrine resistance, precision medicine, real-world evidence

## Abstract

Endocrine-resistant, hormone receptor-positive, and HER2-negative (HR+/HER2-) metastatic breast cancer (mBC) is largely governed by acquired mutations in the estrogen receptor, which promote ligand-independent activation, and by truncal alterations in the PI3K signaling pathway, with a broader range of gene alterations occurring with less prevalence. Circulating tumor DNA (ctDNA)-based technologies are progressively permeating the clinical setting. However, their utility for serial monitoring has been hindered by their significant costs, inter-technique variability, and real-world patient heterogeneity. We interrogated a longitudinal collection of 180 plasma samples from 75 HR+/HER2- mBC patients who progressed or relapsed after exposure to aromatase inhibitors and were subsequently treated with endocrine therapy (ET) by means of highly sensitive and affordable digital PCR and SafeSEQ sequencing. Baseline *PIK3CA* and *TP53* mutations were prognostic of a shorter progression-free survival in our population. Mutant *PIK3CA* was prognostic in the subset of patients receiving fulvestrant monotherapy after progression to a CDK4/6 inhibitor (CDK4/6i)-containing regimen, and its suppression was predictive in a case of long-term benefit with alpelisib. Mutant *ESR1* was prognostic in patients who did not receive concurrent CDK4/6i, an impact influenced by the variant allele frequency, and its early suppression was strongly predictive of efficacy and associated with long-term benefit in the whole cohort. Mutations in *ESR1*, *TP53*, and *KRAS* emerged as putative drivers of acquired resistance. These findings collectively contribute to the characterization of longitudinal ctDNA in real-world cases of HR+/HER2- mBC previously exposed to aromatase inhibitors and support ongoing studies either targeting actionable alterations or leveraging the ultra-sensitive tracking of ctDNA.

## 1. Introduction

Breast cancer (BC) is the most common cancer diagnosis (12% of total; 25% in women) and the leading cause of cancer death in women worldwide, with the majority of cases attributable to the hormone receptor-positive and HER2-negative (HR+/HER2-) subtype [1]. Among patients with metastatic disease (mBC) of this subtype, the emergence of endocrine resistance is virtually universal and associated with progressively inferior responses to subsequent treatments, mainly chemotherapy-based regimens. The advent of molecular actionability and the continuous refinement of blood-based diagnostic and monitoring technologies is reshaping the cancer care continuum and permeating clinical trial design: circulating tumor DNA (ctDNA) dynamics shows considerable promise in providing early response surrogates and may spare unnecessary toxicity and substantiate therapeutic strategies to achieve more durable effects against less-expanded resistant clones [2,3]. An increasing number of clinical trials attempting to leverage molecular vulnerabilities are underway, in which patient selection, stratification, and monitoring are increasingly dependent on PCR or next-generation sequencing (NGS) testing of ctDNA [4,5,6,7,8,9,10,11]. Some of these include the development of PI3K inhibitors (e.g., alpelisib, inavolisib), novel endocrine therapies (e.g., elacestrant), AKT inhibitors (e.g., capivasertib), or ERBB2 inhibitors (e.g., neratinib), as well as the implementation of adaptive strategies where treatment regimens are modified upon ctDNA dynamics (e.g., *ESR1* in PADA-1). More recent methodologies are based on tumor-informed ctDNA tracking, where personalized panels are built from whole exome (WES) or whole genome sequencing (WGS) of tumor samples, and on plasma copy number burden, mutational signatures, and epigenetic changes [12,13,14,15]. Their higher sensitivity or broader coverage have the potential to transform the way clinical decisions are made and pave the way for next-generation clinical trials.

However, the translatability of this growing body of evidence is hampered by limited access to ctDNA technologies in the clinical setting, where the cost-effectiveness of large genomic panels is not yet established and where most molecular data come from the study of archival specimens, usually primary tumors. Such retrospective analyses fail to capture clonal heterogeneity under therapeutic pressure and may introduce artifacts due to prolonged fixation [16,17]. Furthermore, the choice of an adequate technology is challenging as significant unknowns remain regarding inter-technique operating characteristics and bioinformatic processing, the clinical implications of variant allele frequencies, and suitable testing timing. With the aim of capturing the feasibility and utility of serial ctDNA tracking in a real-world population of women with aromatase inhibitor-resistant HR+/HER2- mBC, we conducted a prospective study investigating the prevalence and dynamics of a selected group of gene alterations ranked in the European Society of Medical Oncology (ESMO) Scale for Clinical Actionability (ESCAT) (*ESR1*, *PIK3CA*, *AKT*, *ERBB2*, *KRAS*, and *TP53*) by means of highly sensitive digital PCR and SafeSEQ NGS technologies.

## 2. Results

### 2.1. Baseline Characteristics of the Study Population

From February 2017 to August 2021, 75 patients with HR+/HER- mBC were enrolled in our cohort. All subjects had a blood sample taken at enrolment (baseline, t0), 58 (77%) gave an on-treatment sample (8 ± 2 weeks after start of treatment, t1), and 46 (61%) had a sample at progression (t2). Ninety-five tumor tissue samples were available for analysis from 62 patients. The median age of patients at the time of enrolment was 65 years (range 39–87 years) (Table 1). Most of the patients (81%) had recurrent metastatic disease rather than de novo (19%). The median time from first diagnosis to first evidence of metastasis was 6.0 years (0–10.2), and the median time from first diagnosis to enrollment was 9.1 years (0.3–14.0). Half of the patients (52%) had visceral metastatic disease at enrollment, including involvement of the liver (23%), lung (21%), and pleura (15%). Of note, none of the patients had CNS involvement at baseline, 21% had bone-only disease, and 7% had node-only disease. The number of previous lines of therapy in the metastatic setting was 0 in 16 patients (21%), one in 38 patients (51%), two in 15 patients (20%), and three or more in six patients (8%), with 9 (12%) having received a prior line of non-adjuvant chemotherapy. The median cumulative duration of prior AI therapy in any setting was 28.4 months (1–119.7), with 30% and 28% of patients meeting the criteria of secondary and primary resistance, respectively. Primary resistance was defined as relapse during the first 2 years of adjuvant ET or progressive disease within the first 6 months of first-line ET in the metastatic setting, and secondary resistance was defined as relapse while on adjuvant ET but after 2 years of treatment, relapse within 1 year of completing adjuvant ET, or progressive disease after 6 months of ET in the metastatic setting [18]. Thirty-nine percent of patients had received prior therapy with CDK4/6i. Following enrolment, 71% of patients received SERD-containing regimens, 29% received AI-containing-regimens, and 24% received a concurrent CDK4/6i. The median follow-up of study participants was 29.6 months (range 4.4–66.8); 81% (*n* = 61) of the patients experienced progression of the disease, and 19% (*n* = 14) died. The median PFS on first, second, and third line of therapy was 12.8 (1.6–54.3), 6.9 (1.3–32.1), and 6.7 months (1.6–34.3), respectively, which is consistent with the existing literature considering the absence of CDK4/6i in a substantial proportion of the patients due to the long follow-up in our series. As best response as per RECIST 1.1, 40% of patients attained a partial response, 73% derived clinical benefit, and 27% had progression of the disease.

### 2.2. Mutational Landscape in Baseline ctDNA

All of the patients had t0 plasma samples analyzed with dPCR interrogating the following hotspot mutations: PIK3CA E545K, E542K, and H1047R, and ESR1 Y537S and D538G. Fifty-four patients had their t0 samples analyzed with the SafeSEQ NGS breast cancer panel covering the coding regions of PIK3CA, ESR1, TP53, AKT, ERBB2, and KRAS. Overall, the mutation prevalence at baseline was 44% (33/75) in PIK3CA, 39% (29/75) in ESR1, 20% (11/54) in TP53, 6% (3/54) in AKT, 2% in ERBB2 (1/54), and 0% in KRAS (Figure 1A,B). No significant dependency or exclusion was identified among gene alterations. The most frequent mutation in PIK3CA was E545K (24%), followed by E542K (13%) and H1047R (11%), and the most prevalent in ESR1 was D538G (22%), closely followed by Y537S (16%) (Figure 1C). The median PIK3CA VAF was 0.46% (0.013–44.67%) and the median ESR1 VAF was 0.86% (0.010–25.02%). Four patients (4/75; 5%) exhibited concurrent polyclonal mutations in PIK3CA and 4% (3/75) in ESR1. A significant proportion of patients (21%; 16/76) had coexisting PIK3CA and ESR1 mutations, whereas 40% (30/75) had none. Of the 11 patients with TP53 mutations at t0, six had a PIK3CA mutation, six had an ESR1 mutation, and these coexisted in four patients. One of the three patients with baseline AKT mutations had concurrent PIK3CA and ESR1 mutations, while no alterations coexisted in the only case with an ERBB2 mutation. Overall, these results are consistent with the mutation prevalence reported in larger published cohorts (Figure 1D). A positive association was observed between the prevalence of ESR1 mutations and the cumulative AI exposure time and the number of lines of endocrine therapy prior to t0, and patients with baseline plasma TP53 mutations tended to have a higher histological grade, visceral disease, and history of chemotherapy regimen in the metastatic setting (Appendix A).

### 2.3. Concordance between dPCR and SafeSEQ NGS

Mutational data obtained from both dPCR and the SafeSEQ Breast Cancer panel were available for concordance analysis in 121 samples from 54 patients. A median of 4 mL of plasma were used for dPCR testing, while a median of 2.5 mL (1.9–3 mL) of the remaining plasma were used for the NGS panel. The concordance rates for mutations in PIK3CA and ESR1 were 87% and 86%, respectively, with positive percentage agreements (PPA) of 94% and 85% and negative percentage agreements (NPA) of 84% and 86%, respectively (Figure 2A). Among patients with no mutations detected by dPCR, the expanded coverage of SafeSEQ enabled the detection of clinically relevant mutations in PIK3CA and ESR1 in 10 (8%) and three (2%) samples, respectively. The lowest VAF value observed with dPCR was 0.013%, and the lowest VAF value observed with SafeSEQ was 0.049%. The major reason for the discordant cases, however, were mutations with a VAF < 1% detected by dPCR but undetectable by targeted NGS, where concordance was 43% and 41% for mutations in PIK3CA and ESR1, respectively (Figure 2B). Among concordant cases, a strong correlation was observed between VAF levels (Spearman’s ρ = 0.94 [95% CI 0.89–0.97], *p* < 0.001) (Figure 2C) as well as number of mutant molecules per mL of plasma (ρ = 0.91 [95% CI 0.84–0.95], *p* < 0.001) (Figure 2D). No significant difference between assays was found when comparing concordance rates across samples at different timepoints.

### 2.4. Clonal Evolution from Archival Tissue to Liquid Biopsy

A total of 94 archival tissue samples from 58 patients were available for dPCR testing. Six samples were excluded due to low cellularity (<30%). One sample was analyzed per patient, prioritizing those obtained closest to the baseline plasma timepoint. Twenty-seven samples corresponded to primary tissue, and 31 were biopsies of metastasis. PIK3CA mutation prevalence was 70% in primary, 61% in metastasis, and 39% in plasma samples. Excess was mostly due to mutations with VAF < 0.1%, appearing in 22% of primary samples, 26% of metastatic samples, and 10% of plasma samples (Appendix A). Conversely, ESR1 alterations were observed in 11% of primary, 36% of metastasis, and 32% of plasma samples, with mutations with VAF < 0.1% ranging from 7% to 10%. The detection of lower VAFs was correlated with a higher DNA content (Appendix A). To control for false positives, dPCR interrogating PIK3CA and ESR1 mutations was performed in a cohort of 25 healthy patients, and no mutant copies were detected. Notably, the patients with ESR1 mutations in their primary tumor tissue (*n* = 3) had not received prior AI, one had received tamoxifen, and none had received hormone replacement therapy. No significant enrichment of particular mutations was observed across sample types. The overall concordance of mutational status between tissue and ctDNA was 54% and 68% for PIK3CA and ESR1, respectively, which increased to 72% and 76% if variants with VAF < 0.1% were excluded. While most excess PIK3CA mutations were detected in tissue, discordant ESR1 mutations were balanced between tumor and plasma (Appendix A). Only two of the patients who had a mutation in tumor tissue but not in baseline ctDNA presented an alteration in subsequent time points (t1, t2). Otherwise, concordance was not significantly influenced by the tissue sample type, the time between tumor biopsy and blood draw, or the use of prior CDK4/6i.

### 2.5. Cell-Free DNA at Baseline and Early Dynamics

The putative prognostic and predictive value of cfDNA levels was explored in our cohort. The median number of the sum of PIK3CA and ESR1 copies per microliter at baseline was 256 (range 23–6751). cfDNA levels were significantly higher at the time of progression compared to baseline and early in treatment (Figure 3A). Higher baseline cfDNA levels were associated with a worse prognosis (HR 1.75, 95% CI 1.05–2.90, *p* = 0.02) after adjusting for age, ECOG, and the presence of visceral disease (Figure 3B). Baseline cfDNA levels were not significantly associated with the number of prior lines of therapy, exposure time to endocrine therapy, the presence of visceral disease or bone-only disease, or the prior use of CDK4/6i. The early decrease in cfDNA levels showed a non-significant trend towards an improved PFS and clinical benefit as per RECIST 1.1 (Figure 3C; Appendix A).

### 2.6. Circulating Tumor DNA at Baseline

We next investigated the prognostic value of baseline ctDNA in our cohort. PIK3CA mutations at baseline were associated with a shorter PFS in multivariate analysis adjusting for age, performance status, and the existence of visceral disease (HR 2.14, 95% CI 1.25–3.70, *p* = 0.033) (Figure 4A,B). This prognostic role was statistically significant in patients receiving fulvestrant-based therapies (*n* = 53; HR 2.63, 95% CI 1.32–5.0, *p* = 0.013) but not in those receiving AI-based therapies (*n* = 22; *p* = 0.9) or those receiving CDK4/6i during the study period (*p* = 0.88) (Appendix A). Of note, PIK3CA mutant patients tended to have a longer PFS with exemestane plus everolimus (Appendix A). As a niche of highly unmet need, we observed that PIK3CA mutations were associated with a worse prognosis in the subset of patients who had progressed to a prior CDK4/6i and received fulvestrant monotherapy while on study (*n* = 13; HR 5.49, CI 95% 1.60–19.30, *p* = 0.006) (Figure 4C). ESR1 mutations were significantly prognostic in patients who did not receive CDK4/6i while on study (*p* = 0.049), but this association failed to achieve statistical significance for the whole population (*p* = 0.16) or for specific treatment subgroups including exemestane plus everolimus (*p* = 0.79) (Figure 4D,E; Appendix A). No significant association with a worse outcome was identified with specific ESR1 or PIK3CA mutations.

We next inquired whether the VAF had a relevant impact on prognostic associations. Patients with a baseline ESR1 VAF > 1% performed significantly worse than those with VAF < 1% (Figure 4F). Of note, the latter exhibited an outcome overlapping with non-mutant patients. No significant differences were observed in PIK3CA mutant patients (Appendix A). When stratified by type of therapy, patients with baseline PIK3CA mutation had numerically longer PFS with AI compared to fulvestrant (*p* = 0.08), while patients with ESR1 mutations displayed overlapping curves (*p* = 0.69) (Appendix A).

Of the 54 patients whose baseline samples were interrogated for TP53 alterations, 11 displayed pathogenic mutations, which were associated with a worse PFS (HR 2.4, CI 95% 1.03–6.17, *p* = 0.03) (Figure 4G). The limited sample size precluded us from subgrouping TP53 mutant patients and constrained our ability to ascertain the weighted role of each mutation in patients with more than one mutant gene, although patients with tumors harboring combinations of mutant genes did not exhibit significantly worse outcomes compared to those with individual mutations (Appendix A).

### 2.7. Circulating Tumor DNA Dynamics

The role of early ctDNA dynamics in patients with available t0 and t1 samples (*n* = 58) was investigated. In our series, the early suppression (t0 > t1 = 0) of PIK3CA mutant copies did not predict an improved PFS (*p* = 0.64) (Figure 5A). When stratified by treatment type, we observed that PIK3CA mutant patients on AIs (*n* = 8) who achieved an early suppression of mutant copies had a very prolonged PFS compared with those without suppression (*p* = 0.03), an impact that was not evidenced in those receiving fulvestrant (*n* = 19; *p* = 0.41) (Appendix A). Conversely, the early suppression of mutant ESR1 copies was a strong predictor of a better outcome in the whole cohort (HR 0.13, 95% CI 0.03–0.58, *p* = 0.003), and it remained significant after adjusting by age, ECOG PS, and the presence of visceral disease (Figure 5B). Among patients with ESR1 mutant ctDNA, the early suppression of ESR1 mutant copies was predictive of a better PFS with fulvestrant (*n* = 17; *p* = 0.004) but not AIs (*n* = 8; *p* = 0.45) (Appendix A). Of note, the early suppression of ESR1 mutant copies was predictive of long-term benefit (>9 months) with high sensitivity and negative predictive value, with all the patients without suppression experiencing progression before 9 months of therapy (Figure 5C–E). Finally, the early dynamics of TP53 VAF was not clinically informative (*p* = 0.86), although this analysis was based on a limited number of mutant cases. At the time of radiological progression, PIK3CA mutation prevalence was similar to previous timepoints, while ESR1 (t2 46% vs. t0 38% and t1 29%) and TP53 mutations (t2 28% vs. t0 20% and t1 26%) were enriched.

### 2.8. Individual Cases and Subgroups of Clinical Interest

We finally explored unique cases or subgroups within our cohort that could inform future studies on cfDNA or actionability in HR+/HER- mBC. ctDNA dynamics are illustrated in Figure 6 along with clinical vignettes.


PI3K pathway targeting:


One of our patients (#5; Figure 6A) received fulvestrant and alpelisib as fourth line of therapy. A baseline PIK3CA mutation was identified (E545K; VAF 8.25%). This patient, who had received prior therapy with CDK4/6i, achieved a partial response and had a PFS of 34.3 months. Liquid biopsy monitoring revealed an early PIK3CA VAF suppression on treatment. At radiological progression, the PIK3CA E545K VAF rose to 6.17%, new mutations emerged in ESR1 (Y537S; VAF 0.77%) and TP53 (H193L; VAF 0.047%), and cfDNA increased by almost five times.

Two other patients (#47 and #56; Figure 6B,C) received fulvestrant and capivasertib as second line of therapy after AI plus a CDK4/6i. None of them had PIK3CA mutations at baseline, and #47 had an ESR1 mutation at baseline (D538G; VAF 0.13%). For #47, ESR1 VAF rose continuously to 0.39% at radiological progression after 4.6 months of treatment. For #56, a new PIK3CA mutation (E545K, VAF 1.64%) and an ESR1 mutation (Y537S; VAF 1.45%) emerged at progression at 3 months.

One patient (#38; Figure 6D) received fulvestrant and mTOR inhibitor sapanisertib and had available longitudinal plasma samples presenting concurrent baseline PIK3CA (E545K; VAF 0.22%), ESR1 (D538G; VAF 0.86%), and TP53 (G244S; VAF 0.48%) mutations. With evidence of progressive disease only at 1.8 months of treatment, PIK3CA VAF was suppressed, ESR1 VAF was decreased (VAF 0.64%), TP53 VAF was increased (VAF 0.72%), and cfDNA content was slightly increased.


PIK3CA, ESR1 and TP53 triple mutants:


Four of our patients had concurrent mutations at baseline. Only #48 (Figure 6E), receiving second-line fulvestrant and capivasertib, had a PFS longer than 3 months (9.3 months). At t1, PIK3CA VAF decreased to 24.0% from 44.7%, ESR1 VAF was suppressed, and TP53 was also substantially reduced to 31.3% from 61.7%.


Emerging KRAS mutations:


None of our patients presented with KRAS mutations at baseline, but two of them developed them at progression (Figure 6F,G). #10 received fulvestrant and ribociclib as fourth line of therapy after prior AI, fulvestrant, and exemestane plus everolimus. This patient had a PFS of 10.4 months and SD as best response. Concurrent PIK3CA (E542K; VAF 0.19%) and ESR1 (D538G; VAF 0.16%) mutations were detected at baseline. PIK3CA VAF continuously rose (t1 VAF 0.34%) while ESR1 VAF was suppressed, and both were detectable at progression along with an emerging KRAS mutation (G12C; VAF 0.06%). Patient #18 received fulvestrant and ribociclib as first line of treatment but experienced upfront resistance with a PFS of only 2.5 months. There were baseline mutations in PIK3CA (E545K, VAF 10.63%; C420R and E726K), ESR1 (E380Q; VAF 5.86%), and TP53 (R156S; VAF 3.55%), all of which were present but reduced at evidence of radiological progression together with a new KRAS mutation (G12C; 0.12%).


ERBB2 mutation:


Patient #31 (Figure 6H) was the only one displaying an ERBB2 baseline mutation in our cohort. She received exemestane and everolimus as third line of therapy and had not received prior CDK4/6i. At baseline, concurrent PIK3CA (E545K; VAF 0.013%) and ERBB2 (L313V; VAF 8.17%) were detectable. At t1, PIK3CA VAF was suppressed, ERBB2 VAF increased to 10.96%, and a new TP53 mutation appeared (M340L; VAF 0.07%). No plasma samples were available at the time of radiological progression, with the patient attaining a partial response and a PFS of 16.4 months.


HER2-low versus HER2-zero:


The HER2 immunohistochemistry score was available in 67 (89%) of our patients. Twenty (26%) of them were categorized as HER2-zero and 47 (63%) as HER2-low (defined as IHC 1+, or 2+ and FISH negative) as per ASCO CAP guidelines [24]. No statistically significant differences were observed in the distribution of ESR1, PIK3CA, or TP53 mutations between the two subgroups (Figure 6I).


Clinical outliers:


Five patients within our cohort exhibited remarkably prolonged PFS (>12 months) in the third- or fourth-line setting (Figure 6J). PFS ranged from 15.3 to 34.3 months, and all of them attained a partial response. Two of them had received prior CDK4/6i. Four of them displayed a PIK3CA mutation at baseline, of whom one received fulvestrant and alpelisib and the rest received exemestane and everolimus, and two of them had an ESR1 mutation. At progression, PIK3CA and ESR1 mutations were detectable in two of four cases with available samples, and a new TP53 mutation had emerged in three of them. In all cases, cfDNA levels were increased.

Conversely, six patients receiving first-line treatment and experiencing a cancer-related event had a PFS of less than 6 months (range 2.1 to 5.1 months) (Figure 6K). Four patients were treated with CDK4/6i, and two received fulvestrant alone. Of them, two were PIK3CA mutant, one was ESR1 mutant, and three were TP53 mutant at baseline. At progression, two patients developed new PIK3CA mutations, two additional patients developed new ESR1 mutations, and one patient developed a KRAS mutation.

## 3. Discussion

This study prospectively followed a real-world cohort of HR+/HER2- mBC patients who had received AIs in the adjuvant or metastatic settings and were treated with ET and monitored with serial liquid biopsies. The long clinical follow-up, comprehensive annotation, and use of affordable, highly sensitive dPCR and SafeSEQ technologies provide a unique perspective on the challenges and opportunities of mutational analysis to guide therapeutic decisions in the daily clinic.

Our cohort was composed of postmenopausal women with AI-resistant disease, half of them with visceral involvement and 79% being treated in second or later line of therapy. The landscape of circulating mutations at baseline was similar to published evidence from larger trial populations and cohorts [19,20,21,22,23]. *PIK3CA* mutant prevalence in ctDNA remained similar irrespective of treatment background, plasma sample time, or HER2-low versus zero status. In primary tissue, the rate of *PIK3CA* mutations was unexpectedly high, which could be partially explained by the selection of patients with worse prognosis and the high sensitivity of dPCR, with the usual prevalence resulting from excluding variants with VAF < 0.1%. *ESR1* mutations were enriched in patients with a higher burden of prior endocrine therapy, in ctDNA samples at progression, and in metastatic tissue samples. Similarly, *TP53* mutations were enriched in patients with high grade disease, visceral involvement, and a history of previous chemotherapy in the advanced setting. Recent reports have described mutual exclusivity between *ESR1* and *TP53* mutations in mBC, but we found concurrence in six of the 11 *TP53* mutant patients [25,26]. While the reasons for this remain unclear, our increased sensitivity might have contributed given that *ESR1* VAFs were <1% in five of these cases. The spectrum of mutations was also similar, with the exception of *PIK3CA* E545K being the most frequent alteration instead of H1047R. Again, this mutation was identified in our cohort with VAFs significantly lower, and we hypothesize that it may have been missed in studies that used less sensitive approaches. Additionally, our cohort is more heavily pretreated that the ones analyzed in landmark studies, which may have contributed to emerging subclonality.

With the future possibility of leveraging ctDNA dynamics to switch treatment without evidence of clinical progression, clinicians today require affordable methods that can interrogate clinically meaningful alterations with adequate sensitivity and specificity in all patients. Thus, there is an ongoing practical debate regarding whether to prioritize larger panels versus more sensitive, but narrower, approaches. The clinical context becomes critical where metastatic disease is likely to present with higher circulating tumor loads and enable the detection of actionable/prognostic mutations with sufficient precision and on-treatment monitoring with sufficient sensitivity. Overall, ctDNA was detected in 65% of our population, with no significant associations with disease presentation characteristics. We controlled for false positives by interrogating *PIK3CA* and *ESR1* mutations in a cohort of 25 healthy patients, none of whom exhibited mutant copies. We may thus argue that misdiagnoses of mutation status using our approach were unlikely to have a substantial impact on outcomes in our metastatic cohort. The robust concordance between dPCR and the NGS SafeSEQ panel helps extend the number of informative pathogenic variants that can be interrogated in limited plasma volumes. While concordance was reduced when VAF < 1%, this may be partially related to the lower plasma input deployed for SafeSEQ analysis (median 4 mL vs. 2.5 mL), which in turn allowed for the identification of less frequent *PIK3CA* and *ESR1* mutations in 13 samples as well as prognostic *TP53* mutations or resistance *KRAS* mutations.

Tissue samples were available for the majority of patients to allow for the analysis of mutation status concordance and temporal evolution of clonality. Intriguingly, plasma–tissue concordance was lower than expected [27,28]. The reasons for this are unclear, as we did not identify a significant association between discordance and the sample type (primary versus metastatic), time since tissue biopsy to baseline ctDNA, number of lines of therapy in the interval, prior use of CDK4/6i, or evidence of lower VAFs. Most of the discordance was due to *PIK3CA* mutations detected in tumor tissue but not in ctDNA. Similarly, the overall concordance between plasma and tissue testing in SOLAR-1 was 71.5%, with discordance mostly attributed to tissue-only detection. Moreover, in a comparable population, Suppan et al. recently reported an overall concordance rate of 72.2% (52/72) and observed lower VAFs in discordant cases [29]. We may hypothesize that our limited numbers and heterogenous history of previous treatments may have impacted the above-mentioned contrasts. This is supported by the observation that only two of the patients with tumor tissue but no baseline ctDNA mutations presented that same alteration in subsequent timepoints. Also of note, three of 27 (11%) primary tumor samples from patients not exposed to AI displayed an *ESR1* mutation. While we would expect this prevalence to be around 1–7%, both the available number of tissue samples, higher sensitivity, and the fact that our population was composed of women with AI-resistant tumors who developed metastatic disease could reasonably explain the higher frequency [30,31,32].

The role of cfDNA as a prognostic biomarker is well known [33,34]. Here we combined *PIK3CA* and *ESR1* copies to generate an accessible estimate of cfDNA levels available from all of our patients. We demonstrated that baseline cfDNA levels were independently prognostic, and likely only the limited sample size precluded us from identifying a statistically significant predictive role. This and similar simple mutation agnostic approaches may help select those patients who can benefit from larger, more expensive panels, and therefore earlier interventions.

Baseline *PIK3CA* and *TP53* mutations were strongly prognostic in the whole cohort. *ESR1* mutations, however, were prognostic only in patients not receiving CDK4/6i. The observation that CDK4/6i overcome the adverse effect of *ESR1* mutations has already been described [11,20,35,36]. Moreover, at the time of radiological progression, *ESR1*, but not *PIK3CA* mutations, were enriched, which may point to its role as a driver of acquired resistance [37,38]. These findings align with mounting evidence of the activity of the new generation of SERDs irrespective of *ESR1* status and support their development in the AI-resistant setting [39,40]. Also of note, *PIK3CA* mutations were prognostic in patients treated with fulvestrant after progression to CDK4/6i, a subgroup of a worse outcome [39]. Neither *PIK3CA* nor *ESR1* mutations significantly predicted the efficacy of exemestane plus everolimus, although a trend towards a better outcome was evident in *PIK3CA* mutant patients. While this is consistent with previous studies, the existing data are conflicting [41,42,43]. Interestingly, baseline *ESR1* VAFs > 1% were associated with a worse outcome than those <1%, while this was not recapitulated for *PIK3CA* mutations. This finding supports the use of sensitive approaches with sufficient sequencing depth over other technologies (e.g., WGS, WES) when the goal is to characterize mutations with actionable potential that can be acquired, subclonal, and varying under therapeutic pressure. When interrogating early ctDNA dynamics, the suppression of mutant *ESR1* copies was a strong predictor of an improved outcome [44,45,46]. Particularly, *ESR1* suppression was associated with a higher likelihood of long-term benefit (PFS > 9 months).

We leveraged the extensive annotation and prolonged follow-up time of our cohort to explore illustrative clinical cases and subgroups. One of them exemplified the predictive role of early *PIK3CA* suppression in a *PIK3CA* mutant patient receiving fulvestrant and alpelisib after progression to a CDK4/6i, consistent with the published results from the BYLieve trial [6]. None of our patients treated with capivasertib had a *PIK3CA* mutation at baseline, thus precluding any comparison with the recent updates from the FAKTION and CAPItello-291 trials where an altered PI3K pathway conferred greater benefit [5,47]. We also observed the emergence of *KRAS* mutations as drivers of acquired resistance in two patients receiving fulvestrant and ribociclib, which adds to emerging evidence pointing to KRAS mutations as valuable ctDNA biomarkers of CDK4/6 resistance and provides a potential actionability in light of the recent advances in RAS druggability [48,49].

Our results are hypothesis-generating, in line with ongoing clinical research, and provide insight into the applicability of plasma monitoring of well-characterized gene alterations, namely *PIK3CA* and *ESR1*, in a real-world environment. This study has several limitations. First, a relatively small number of patients was involved, thus precluding subgroup comparison and hindering statistically significant associations. Second, our population had received a heterogenous number of treatment lines before enrollment, and due to the long follow-up, most of them did not receive upfront CDK4/6i. Additionally, most of the patients treated with AI did also receive everolimus. Third, while dPCR data was available for all samples, SafeSEQ was performed only in patients who had remaining plasma after dPCR, thus reducing the analyzed population and possibly limiting the detection of lower VAFs.

Liquid biopsy shows clear advantages over tumor tissue analysis such as providing real-time information on treatment response, which is especially relevant in metastatic patients. However, it is still not a standard clinical tool in the analysis of predictive biomarkers partly due to its limited sensitivity and precision and the lack of standardized methods. Not only is ctDNA 0.1–10% of total cfDNA, but results may also depend on the size, stage, and location of the disease. Therefore, more efforts are required to enhance the operational characteristics of liquid biopsy in order to reduce false-positive and false-negative cases. In our study, we used highly sensitive and accessible methodologies, including an NGS panel that increase the information obtained from the tumor. Future investigations leveraging other components of the tumor circulome such cfRNA or methylation changes are underway, as well as enrichment strategies using extracellular vesicles and bioinformatic tools that measure tumor fraction and integrate clinical variables, improving the downstream analysis.

## 4. Materials and Methods

### 4.1. Study Design and Patient Cohort

A prospective observational study was conducted in patients with HR+/HER- mBC who had been previously treated with aromatase inhibitors (AI) in either the adjuvant or the metastatic setting and were scheduled to receive endocrine therapy (ET) with an AI or a selective estrogen receptor degrader (SERD) with or without CDK4/6 inhibitors (CDK4/6i) or other targeted agents. The addition of CDK4/6i to ET was approved in Spain for this indication in November 2017. Patients were consecutively enrolled and treated at the Breast Cancer Unit of the Hospital Clinico San Carlos (HCSC), a tertiary university hospital providing care for a large urban population with nearly 500,000 inhabitants. Major inclusion criteria also included an Eastern Cooperative Oncology Group (ECOG) scale performance status of 0 to 1, an absence of symptomatic visceral involvement, and a life expectancy of more than 6 months. Five enrolled patients were excluded for the following reasons: no prior AI exposure (*n* = 2), baseline sample collected 1 week after start of treatment (*n* = 1), new ET line without AI or SERD (*n* = 2). Plasma analysis was performed retrospectively and thus did not inform patient management while on study. This study was approved by the Institutional Ethical Committee of HCSC (16/087-E_BD) and conducted in accordance with Good Clinical Practice Guidelines and the Declaration of Helsinki. All participants provided written informed consent before participation.

### 4.2. DNA Extraction

Approximately 20 ml of blood was drawn and collected in EDTA tubes at baseline (t0), at 8 ± 2 weeks on treatment (t1), and at disease progression (t2). Within 2 h of collection, blood samples were centrifuged at 1900× *g* for 10 min. The plasma obtained was centrifuged at 16,000× *g* for 10 min to remove cell debris and aliquoted and stored at −80 °C until DNA extraction. Circulating DNA was isolated from plasma using the QIAamp Circulating Nucleic Acid Kit (Qiagen, Hilden, Germany). This protocol was modified extending up to 1 hour the incubation time at 60 °C. Blood was also used to obtain germline DNA (gDNA) from peripheral blood mononuclear cells using the MagNA Pure Compact Nucleic Acid Isolation Kit (Roche Diagnostics, Grenzach-Whylen, Germany) to test germline and clonal haematopoiesis mutations. Regarding tumor samples, 4–8 sections of paraffin-embedded tumor tissue were used for DNA extraction by the GeneRead DNA FFPE Kit (Qiagen), following the manufacturer’s instructions. The tumor region was selected by a pathologist, and samples with <30% tumor cells were excluded. Isolated DNA was quantified by Qubit 3.0 Fluorometer (Thermo Fisher Sci, Waltham, MA, USA).

### 4.3. Detection of ESR1 and PIK3CA Mutations by Digital PCR (dPCR)

Hotspot mutations in *PIK3CA* (E545K, E542K, and H1047R) and *ESR1* (Y537S and D538G) were analyzed in cfDNA isolated from 4 ml plasma using QuantStudio 3D Digital PCR System. Wet lab-validated TaqMan dPCR Liquid Biopsy assays were used for PIK3CA analysis (ThermoFisher Sci, Waltham, MA, USA). TaqMan probes were designed for ESR1 analysis using Custom TaqMan Assay Design Tool (ThermoFisher Sci) (Appendix A). Data analysis based on FAM (probe used for mutant) and VIC (probe used for wild-type) events were performed using QuantStudio 3D AnalysisSuite Cloud Software (version 3.1.6; ThermoFisher Sci). A quality threshold of 0.6 was applied. The concordant result of two measurements for each experiment was necessary to consider the presence of mutation. The variant allele fraction (VAF; percentage of mutant ctDNA copies relative to the total cfDNA) and number of mutant molecules per mL were reported and analyzed. Mutations detected in cfDNA with VAF > 40% or <1% were also analyzed in gDNA to test germline or clonal haematopoiesis mutations, respectively. No germline nor clonal haematopoiesis mutations were identified.

### 4.4. NGS SafeSEQ Targeted Panel

Circulating free DNA (cfDNA) isolated from plasma (1.9–3 mL) from 54 patients was also subjected to mutational analyses with a focused breast cancer panel assay (BC-SEQ), based on the Safe Sequencing System (SafeSEQ; Sysmex Inostics, Hamburg, Germany) [50]. The BC-SEQ assay uses unique molecular identifiers, enabling detection and accurate quantification at Variant Allele Frequencies (VAFs) as low as six mutant molecules in 20,000 circulating free DNA genomic equivalents (0.03%). BC-SEQ detects single-nucleotide variations (SNVs), insertions (up to nine nucleotides), deletions (up to 24 nucleotides), and deletion/insertion variants (up to 16 nucleotides) in six genes (*ESR1*, *PIK3CA*, *ERBB2*, *AKT1*, and *TP53*), using a total of 28 amplicons. The regions within these six genes in which mutations were detected using this assay are shown in Appendix A.

Plasma samples were frozen at −80 degrees C and shipped in batches to the Sysmex Inostics laboratory in Baltimore, MD, USA where the CLIA-validated BC-SEQ assay workflow was performed. In brief, DNA was quantified using the LINE-1 method, and between 3000 and 20,000 genomic equivalents were assayed; cfDNA inputs between 4.3 and 86 ng were used for library preparation. Library quality and quantity were checked using an Agilent 2100 Bioanalyzer (Agilent Technologies, Santa Clara, CA, USA). The enriched libraries were sequenced using the NextSeq 500/550 Mid Output Kit v2.5 (150 Cycles) (Library input: 1.0 pM; single-read) on Illumina’s NextSeq 550 platform. The FastQ files were analyzed using a proprietary in-house software developed by Sysmex Inostics Inc. in Baltimore MD (v2.0). The threshold for positive mutation calls for each position was determined based on a panel-wide Limit of Detection (LOD95)-based threshold that applies to mutations identified in all 28 amplicons contained within the BC-SEQ panel. An absolute cut-off of 6 mutant molecules was set as the threshold of positivity for all mutations in the panel; this corresponds to a Variant Allele Frequency (VAF) of 0.03% (6 mutant molecules in a background of 20,000 genomic equivalents or 66 nanograms of total circulating free DNA in plasma). The description and clinical significance of detected mutations was evaluated using ClinVar (Appendix A) and, specifically for *TP53*, the IARC TP53 database ((R20, July 2019): https://tp53.isb-cgc.org; accessed date: 2 November 2022) [51]. Mutations detected in cfDNA with VAF > 40% or <1% were also analyzed in gDNA to test germline or clonal haematopoiesis mutations, respectively. Only *TP53* polymorphism P72R (clinical significance: benign) was detected in gDNA. No mutations based on clonal haematopoiesis were identified.

### 4.5. Statistical Analysis

Patient baseline characteristics including clinical, cfDNA, and ctDNA variables were compared using the Chi-square, Fisher’s exact, Wilcoxon rank sum or Kruskal Wallis tests where appropriate. Progression-free survival (PFS) was defined as the time from treatment initiation to progression or death from any cause, whichever occurred first. Spearman correlation, percent agreement, and Cohen’s kappa coefficients were used to analyze the relationship between dPCR and SafeSEQ results. Survival outcomes were estimated using the Kaplan–Meier method and the Mantel´s log-rank test. Multivariable analysis was performed using the Cox proportional hazards model. Reported effect size and *p* values are bivariate unless otherwise specified. Two-tailed *p*-values <0.05 were considered statistically significant. *p*-values showed in figures: *, *p* ≤ 0.05; ***, *p* ≤ 0.005; and NS, not significant. Statistical analyses and graph design were performed using the R software version 3.9 (https://cran.r-project.org/) and the GraphPad Prism software version 8 (San Diego, CA, USA). Oncoprint and Lollipop plots were generated using OncoPrinter and MutationMapper tools from cBioportal [52,53].

## 5. Conclusions

Our study showed that higher baseline levels of cfDNA and mutations in *PIK3CA*, *ESR1*, and *TP53* were associated with a worse clinical efficacy of endocrine therapy in HR+/HER2- mBC previously exposed to AIs. Baseline cfDNA levels and mutant *PIK3CA* and *TP53* were prognostic. Particularly, *PIK3CA* mutations were prognostic in patients receiving fulvestrant monotherapy after progression to a CDK4/6i-containing regimen, and its suppression was predictive in cases of long-term benefit with alpelisib. Mutant *ESR1* was prognostic in patients not receiving CDK4/6i, and its early suppression was strongly predictive of efficacy and associated with long-term benefit in the whole cohort. The negative impact of *ESR1* on PFS was influenced by the VAF. Mutations in *ESR1*, *TP53*, and *KRAS* emerged as putative drivers of acquired resistance. These findings provide a real-world perspective of the use of serial liquid biopsy monitoring in AI-exposed HR+/HER2-mBC patients and inform the design of novel strategies of patient follow-up and early intervention.

## Figures and Tables

**Figure 1 ijms-24-11419-f001:**
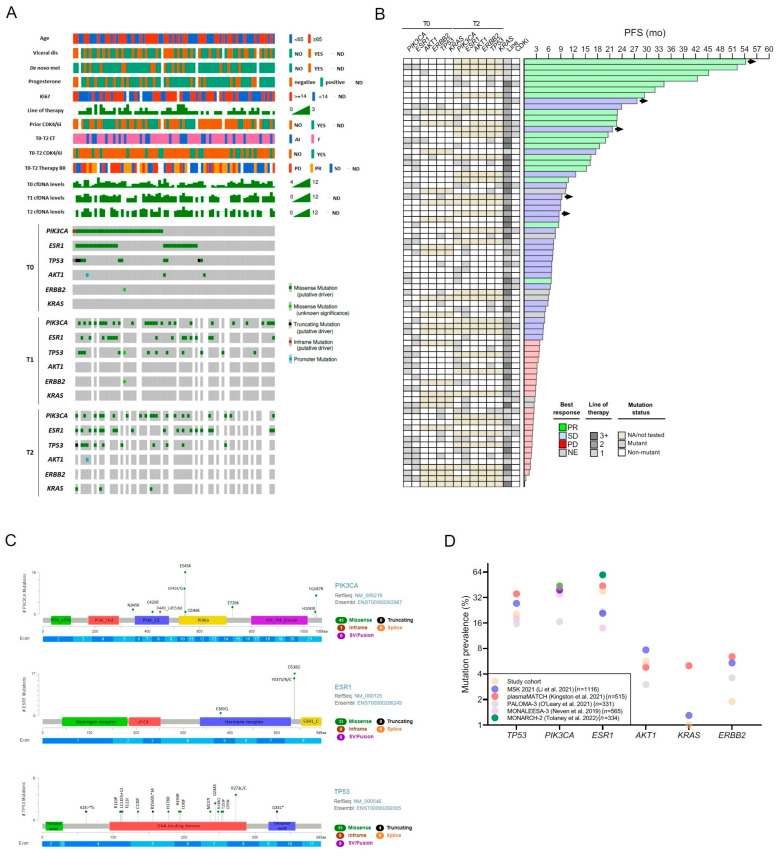
Mutational landscape in baseline ctDNA. (**A**) Oncoprint plot show somatic alterations and clinical–pathological characteristics per patient. (**B**) Swimmer plot displaying individual progression-free survival and mutations in plasma at baseline and at progression. (**C**) Lolliplots showing the spectrum of detected mutations in PIK3CA, ESR1, and TP53. (**D**) Mutation prevalence across selected published studies including landmark trials and large patient cohorts. Abbreviations: PR, partial response; SD, stable disease; PD, progressive disease; NE, non-evaluable; #, number of mutations; arrows, patients with ongoing treatment; [19,20,21,22,23].

**Figure 2 ijms-24-11419-f002:**
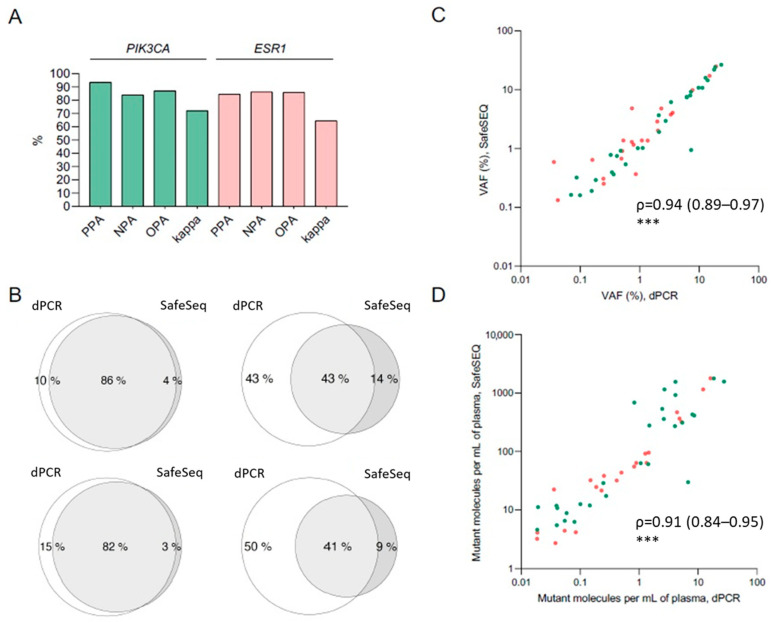
Concordance between dPCR and SafeSEQ NGS. (**A**) Inter-technique agreement in the detection of PIK3CA (green) and ESR1 (pink) mutations. (**B**) Venn diagrams displaying concordance for all analyzed samples (upper left, PIK3CA; down left, ESR1) and mutations with VAF < 1% (right). (**C**,**D**) Correlation of VAF and mutant molecules per mL in mutant samples. Green and red dots denote mutations in PIK3CA and ESR1, respectively. ***, *p* < 0.001.

**Figure 3 ijms-24-11419-f003:**
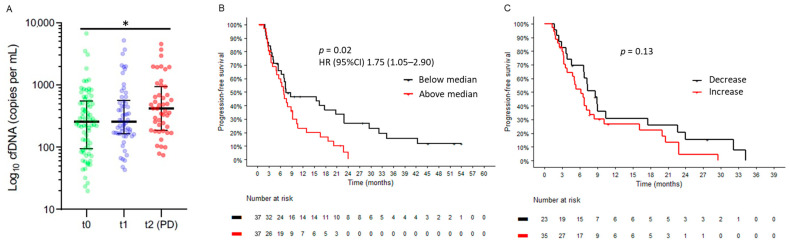
cfDNA levels at baseline and early dynamics. (**A**) Increase in cfDNA levels at progression. (**B**,**C**) Prognostic and predictive value of baseline cfDNA levels. *, *p* < 0.05.

**Figure 4 ijms-24-11419-f004:**
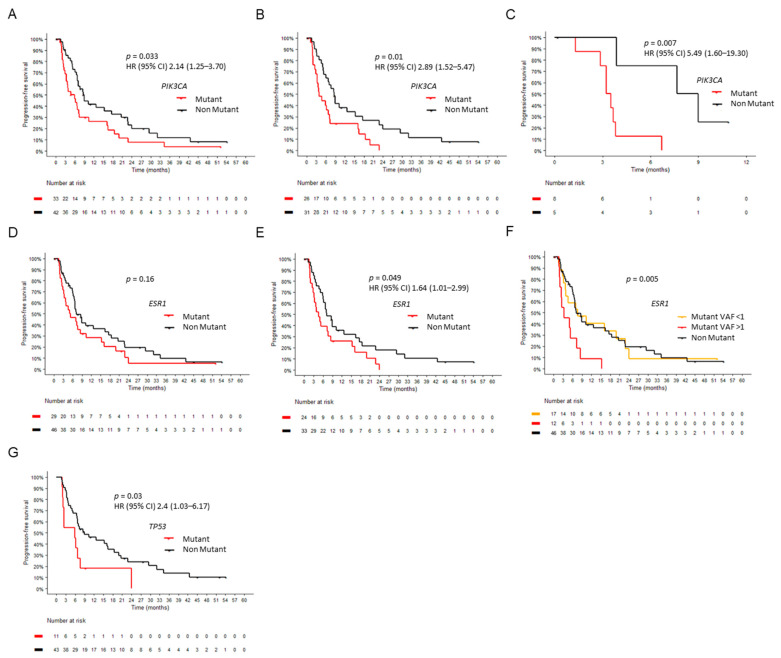
ctDNA levels at baseline. (**A**–**C**) Prognostic value of mutant PIK3CA in the whole cohort (**A**), patients who did not receive CDK4/6i while on study (**B**) and patients who had progressed to prior CDK4/6i and received fulvestrant monotherapy while on study (**C**). (**D**,**E**) Prognostic value of mutant ESR1 in the whole cohort (**D**) and in patients who did not receive CDK4/6i while on study (**E**). (**F**) Differential impact of ESR1 VAF on PFS. (**G**) Prognostic value of TP53 mutations.

**Figure 5 ijms-24-11419-f005:**
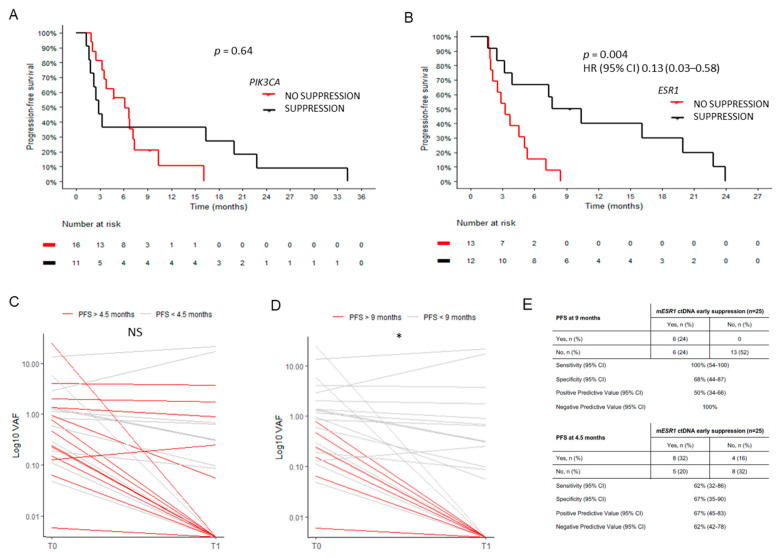
ctDNA early dynamics. (**A**) Lack of a predictive role by early PIK3CA mutant VAF suppression (VAF T1 = 0). (**B**) ESR1 suppression is associated with an improved PFS. (**C**,**D**) ESR1 mutation dynamics from baseline (t0) to 8 ± 2 weeks into treatment (t1) to distinguish patients with upfront resistance (PFS < 4.5 months) or long-term benefit (PFS > 9 months). (**E**) Operational characteristics of ESR1 mutant VAF suppression at 9 and 4.5 months of treatment. NS, not significant; *, *p* < 0.05.

**Figure 6 ijms-24-11419-f006:**
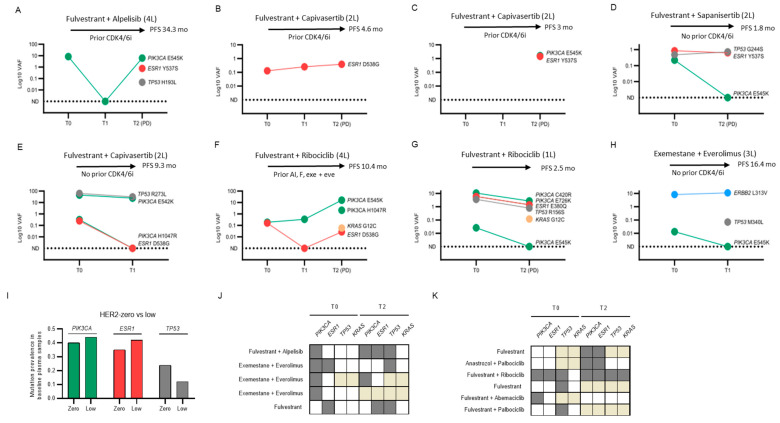
Clinical cases and subgroups of interest. (**A**–**H**) Clonal evolution under therapeutic pressure along with clinical vignettes showing the therapy received during the study period, the previous use of CDK4/6i, and the PFS attained. (**I**) The prevalence of *PIK3CA*, ESR1 (red), and TP53 (gray) alterations stratified by HER2-low or HER2-zero status in our cohort. (**J**,**K**) Grid displaying the therapy received during the study period and the mutation status of PIK3CA, ESR1, TP53, and KRAS at baseline and at progression in patients selected by their unexpectedly prolonged (**J**) or short (**K**) PFS. Shadowing denotes mutant status, white denotes non-mutant status, and gold denotes untested. Green: PIK3CA, Red: ESR1; Gray: TP53; Blue: ERBB2; Orange: KRAS.

**Table 1 ijms-24-11419-t001:** Patient characteristics ^1^.

	All Patients (*n* = 75)	*PIK3CA* Mutant (*n* = 33)	*PIK3CA* Wild-Type (*n* = 42)	*p* Value	*ESR1* Mutant (*n* = 29)	*ESR1* Wild-Type (*n* = 46)	*p* Value
**Age—median (range) yr**	65.0 (39.2–87.7)	66.6 (41.7–87.7)	64.3 (39.2–86.6)	0.87	60.2 (39.2–87.7)	65.5 (39.5–87.1)	0.34
≤65 yr—no. (%)	37 (49)	16 (48)	21 (50)		16 (55)	21 (46)	
>65 yr—no. (%)	38 (51)	17 (52)	21 (50)	0.89	13 (45)	25 (54)	0.42
**Line of therapy—no. (%)**							
First	16 (21)	6 (18)	10 (24)		4 (14)	12 (26)	
Second	38 (51)	16 (49)	22 (52)		13 (45)	25 (54)	
Third or higher	21 (28)	11 (33)	10 (24)	0.63	12 (41)	9 (20)	0.1
**Previous line of chemotherapy—no. (%)**							
Yes	9 (12)	1 (3)	8 (19)		5 (17)	4 (9)	
No	66 (88)	32 (97)	34 (81)	0.07	24 (83)	42 (91)	0.27
**Previous CDK4/6 inhibitor**							
Yes	29 (39)	16 (48)	13 (31)		14 (48)	15 (33)	
No	46 (61)	17 (52)	29 (69)	0.15	15 (52)	31 (67)	0.18
**ECOG-PS—no. (%)**							
0	65 (87)	28 (85)	37 (88)		25 (86)	40 (87)	
1	10 (23)	5 (15)	5 (12)	0.74	4 (14)	6 (13)	0.99
**Endocrine therapy—no. (%)**							
Aromatase inhibitor	22 (29)	10 (30)	12 (29)		9 (31)	13 (28)	
Fulvestrant	53 (71)	23 (70)	30 (71)	0.87	20 (69)	33 (72)	0.79
**CDK4/6 inhibitor—no. (%)**							
Palbociclib	11 (15)	2 (6)	9 (21)		2 (7)	9 (19)	
Ribociclib	5 (7)	3 (9)	2 (5)		4 (14)	1 (2)	
Abemaciclib	2 (3)	2 (6)	0 (0)	0.05	0 (0)	2 (4)	0.03
**Metastatic sites—no. (%)**							
Visceral	39 (52)	13 (39)	26 (62)		19 (65)	20 (43)	
Non visceral	36 (48)	20 (61)	16 (38)		10 (35)	26 (57)	
Bone-only	16 (21)	9 (27)	5 (12)	0.06	5 (17)	5 (11)	0.14
**De novo metastases—no. (%)**							
Yes	14 (19)	7 (21)	7 (17)		7 (24)	7 (15)	
No	61 (81)	26 (79)	35 (83)	0.77	22 (76)	39 (85)	0.33
**Adjuvant endocrine resistance—no. (%) ***							
Sensitive	24 (32)	10 (30)	14 (33)		7 (24)	17 (37)	
Primary	16 (21)	6 (18)	10 (24)		5 (17)	11 (24)	
Secondary	16 (21)	7 (21)	9 (21)		10 (35)	6 (13)	
N/A	14 (19)	7 (21)	7 (17)		7 (24)	7 (15)	
Unknown	5 (7)	3 (10)	2 (5)	0.93	0(0)	5 (11)	0.08
**Primary tumor histology—no. (%)**							
Ductal	34 (45)	10 (30)	24 (57)		12 (41)	22 (48)	
Lobular	9 (12)	5 (15)	4 (10)		5 (17)	4 (9)	
Other (mixed, NOS)	17 (23)	10 (30)	7 (17)		4 (14)	13 (28)	
Unknown	15 (20)	8 (24)	7 (17)	0.09	8 (28)	7 (15)	0.27
**Primary tumor grade—no. (%)**							
Low	4 (5)	3 (10)	1 (2)		1 (3)	3 (7)	
Intermediate	38 (51)	15 (46)	23 (55)		16 (55)	22 (48)	
High	14 (19)	6 (18)	8 (19)		4 (14)	10 (22)	
Unknown	19 (25)	9 (27)	10 (24)	0.39	8 (28)	11 (24)	0.58
**Primary tumor Ki67 (%)—no. (%)**							
1–14	31 (41)	16 (48)	15 (36)		11 (38)	20 (43)	
15–20	10 (13)	5 (15)	5 (12)		5 (17)	5 (11)	
>20	17 (23)	6 (18)	11 (26)		5 (17)	12 (26)	
Unknown	17 (23)	6 (18)	11 (26)	0.54	8 (28)	9 (20)	0.58
**HER2 IHC—no. (%)**							
Low	47 (63)	21 (63)	26 (62)		20 (69)	27 (59)	
Zero	20 (26)	8 (24)	12 (28)		7 (24)	13 (28)	
Unknown	8 (11)	4 (13)	4 (9)	0.72	2 (7)	6 (13)	0.56

Abbreviations: ECOG PS: Eastern Cooperative Oncology Group Performance Status. IHC: immunohistochemistry. N/A: not available. * As per ESMO criteria^14^. ^1^ Unknown values were not computed in statistical analyses.

## Data Availability

The data generated and analyzed in this study are described in the article or its supplementary information documents. Anonymized individual genomic and clinical data can be made available for academic use only upon reasonable request to the corresponding author, under a data transfer agreement, and upon Ethics Committee approval.

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
