# Peer review of "Real-World Use of Highly Sensitive Liquid Biopsy Monitoring in Metastatic Breast Cancer Patients Treated with Endocrine Agents after Exposure to Aromatase Inhibitors"

_ijms, 2023, doi:10.3390/ijms241411419_

Round 1

Reviewer 1 Report

The authors interrogated a longitudinal collection of plasma samples from 75 HR+/HER2- mBC patients who progressed or relapsed after exposure to aromatase inhibitors and were subsequentially treated with endocrine therapy (ET) by means of highly sensitive and affordable digital  PCR and SafeSEQ sequencing. The findings provide insight into serial liquid biopsy monitoring in AI-exposed HR+/HER2- mBC patients and provide insights for designing novel strategies for patient follow-up and early intervention. The stats were properly done and comprehensive survival and expression analysis was done. However, I have minor concerns:

- related work section should be added (as a section or paragraph in the introduction) to give the reader comprehensive view for the current directions. I suggest to highlight recent BC prognostics identification models for survival such as PMID: 35205681 and/or treatment guidance such as PMID: 35205681. references 4-11 should be highlighted with more details in this section. 

- go enrichment and pathway analysis (preferably) could be used to verify the finding from the side of the pathways. 

-some minor English editing, I suggest to redone by a professional editor:

* Collectively, these findings =>  These findings Collectively 

*patients progressed => patients who progressed

* clinical setting but their =>  clinical setting. However, their

Author Response

RESPONSE TO REVIEWER 1 COMMENTS:

Reviewer 1 had a favorable impression of the overall the study, but also suggested that a few changes be made to the introduction/background section. The reviewer also made some editorial suggestions on changing the English sentence structure. We thus reviewed the paper and altered locutions whenever these were located.

Overall, we are grateful for the suggestions for improvements by Reviewer 1.

Comments and Suggestions for Authors

The authors interrogated a longitudinal collection of plasma samples from 75 HR+/HER2- mBC patients who progressed or relapsed after exposure to aromatase inhibitors and were subsequentially treated with endocrine therapy (ET) by means of highly sensitive and affordable digital  PCR and SafeSEQ sequencing. The findings provide insight into serial liquid biopsy monitoring in AI-exposed HR+/HER2- mBC patients and provide insights for designing novel strategies for patient follow-up and early intervention. The stats were properly done and comprehensive survival and expression analysis was done. However, I have minor concerns:

  1. Related work section should be added (as a section or paragraph in the introduction) to give the reader comprehensive view for the current directions. I suggest to highlight recent BC prognostics identification models for survival such as PMID: 35205681 and/or treatment guidance such as PMID: 35205681. references 4-11 should be highlighted with more details in this section.

A brief comment has been included in the introduction as suggested (lines 62-71) to help readers gain context of the current use of liquid biopsy technology and future directions in metastatic HR+ breast cancer. References 12-15 have been also added.

We agree prognostic models have revolutionized the management of early breast cancer and liquid biopsy data will certainly require integration with other data platforms and clinical algorithms. However, we feel an adequate discussion of these concepts falls beyond the scope of this article, where ctDNA detection of well-recognized actionable/prognostic mutations is studied in the metastatic setting, and where the focus on feasibility is central rather than validation or discovery.

  1. go enrichment and pathway analysis (preferably) could be used to verify the finding from the side of the pathways.

We agree that pathway analysis, particularly integrating transcriptomic data, would be interesting to understand the biological processes leading to clinical outcomes.

However, the focus of our work was the study of the dynamics and feasibility of clinical implementation rather than the analysis of the biological underpinnings of specific point mutations, which have been extensively addressed elsewhere in preclinical models and data repositories.

Our paper did focus on a few ESMO-ESCAT recognized actionable and/or prognostic alterations related to breast cancer. We feel the number of genes studied precludes a robust pathway analysis and no transcriptomic data was available from this cohort. In this regard, circulating transcriptomic data technologies still present relevant uncertainties.

That said, efforts to integrate transcriptomics and epigenomics in tissue are underway as part of a larger cohort. And circulating RNA technologies will likely add to patient monitoring and drug discovery (eg. ESR1 fusions, etc) in the short-term, with implementation in several commercial assays being underway.

  1. Comments on the Quality of English Language

-some minor English editing, I suggest to redone by a professional editor:

* Collectively, these findings =>  These findings Collectively

*patients progressed => patients who progressed

* clinical setting but their =>  clinical setting. However, their

Amended in the text as indicated. Moreover, the text has been reviewed for language quality by native authors with extensive experience in high-impact publishing

Reviewer 2 Report

To the authors of the manuscript: “Real world use of highly sensitive liquid biopsy monitoring in metastatic breast cancer patients treated with endocrine agents after exposure to aromatase inhibitors”.

Liquid biopsy may be a new strategy for improving clinical practice in cases of metastatic breast cancer, especially in several main aspects such as the management of metastatic tumor heterogeneity, early prediction of relapse, prediction of response to treatment and even have prognostic value. It is important that there are more studies like the one presented by the authors. That it be prospective and focused on metastatic tumors. Although the results are encouraging, it is true that there is a bias due to the limitation of the sample and the mutations found, as well as the fact that the patients have undergone various treatments.

There are still important aspects to be resolved, that maybe the authors can comment in the discussion, such as the need for greater precision in the results obtained, to reduce "false positives". The results also depend on the type and size of the tumor and the stage. In large tumors and/or with metastases it is easier to detect circulating tumor cells. However, in small or metastatic lesions this detection is more difficult, which is why “false negatives” appear. In addition, it would be necessary to integrate the data obtained with liquid biopsy with other data platforms and analysis algorithms to identify which patients it would be applicable to or which patients at risk could benefit as it may not be feasible for all cases/treatments.

I highlight some minor points:

• Please, I recommend putting asterisks that represent the p value in the figures.

• Figure 5E is not described.

• Regarding figure 6I. The authors comment that they have categorized HER2-low samples that express it as low grade (1) or medium grade (2), both with low expression intensity (+), as well as being negative in FISH. Could you clarify this categorization? A 2+ sample would indicate that there is indeed expression in a significant % of cells in the sample. Is then when the result of FISH is used? Are both results taken into account? Why was a FISH performed?

Author Response

RESPONSE TO REVIEWER 2 COMMENTS:

Comments and Suggestions for Authors

Liquid biopsy may be a new strategy for improving clinical practice in cases of metastatic breast cancer, especially in several main aspects such as the management of metastatic tumor heterogeneity, early prediction of relapse, prediction of response to treatment and even have prognostic value. It is important that there are more studies like the one presented by the authors. That it be prospective and focused on metastatic tumors. Although the results are encouraging, it is true that there is a bias due to the limitation of the sample and the mutations found, as well as the fact that the patients have undergone various treatments.

We appreciate the favorable comments by Reviewer 2 that the manuscript highlighted that: “Liquid biopsy may be a new strategy for improving clinical practice in cases of metastatic breast cancer, especially in several main aspects such as the management of metastatic tumor heterogeneity, early prediction of relapse, prediction of response to treatment and even have prognostic value.” As well Reviewer 2 noted that: “It is important that there are more studies like the one presented by the authors. That it be prospective and focused on metastatic tumors.”

  1. There are still important aspects to be resolved, that maybe the authors can comment in the discussion, such as the need for greater precision in the results obtained, to reduce "false positives". The results also depend on the type and size of the tumor and the stage. In large tumors and/or with metastases it is easier to detect circulating tumor cells. However, in small or metastatic lesions this detection is more difficult, which is why “false negatives” appear.

We agree is an important question. The problem of liquid biopsy monitoring in the metastatic setting is rather the false negative rate given LOD of techniques, mentioned disease characteristics, and treatment effect.

In our cohort, 52% had visceral involvement, 21% had bone-only disease, which is representative of published trials and series. ctDNA was detected at baseline in 65% of the patients, considering all of our population, with no significant associations with disease presentation characteristics. Similarly, baseline cfDNA levels were not significantly associated with the number of prior lines of therapy, exposure time to endocrine therapy, presence of visceral disease or bone-only disease, or prior use of CDK4/6i. Unfortunately, volumetric/sum of diameter data was not available in our population for further adjustment. Overall, we may argue that false negatives of the gene mutations detected here unlikely impacted clinical outcome results.

Regarding false positives, we showed that the detection of lower VAFs was correlated with a higher DNA content (Supplementary Fig. 2C). To control for false positives, dPCR interrogating PIK3CA and ESR1 mutations was performed in a cohort of 25 healthy patients and no mutant copies were detected. Particularly, ESR1 muts may be more influenced given lower VAFs, subclonality, and our observation that lower ESR1 VAFs did not convey a worse outcome.

In response to these suggestions for improvement in the discussion section, we made changes to the discussion to highlight “the importance of understanding the contexts of patient testing where clinical false positive or negative liquid biopsy results might arise” – and where monitoring of mutations detected in plasma with a highly sensitivity method would be very useful to examine therapy efficacy; similarly a method shouldn’t be too sensitive in a clinical context where a sufficient tumor load as determined by liquid biopsy should be present to alert treatment with a systemic therapy.”

To address these points, we’ve added additional discussion: lines 396-404, 451-454, and 481-493 in the revised version of the manuscript.

  1. In addition, it would be necessary to integrate the data obtained with liquid biopsy with other data platforms and analysis algorithms to identify which patients it would be applicable to or which patients at risk could benefit as it may not be feasible for all cases/treatments.

We agree with Reviewer 2’s comment. While this set of actionable/prognostic mutations has provided solid grounds for drug development (i.e. alpelisib, novel SERDs, capivasertib, neratinib), the future of liquid biopsy is constantly reshaping with the integration with other data platforms and analysis algorithms to better identify patients who can derive benefit. Future circulating biomarker data will likely integrate RNA, epigenomics and bespoke methodologies. However, this is still far from becoming a reality in the clinical setting and real-world approaches like ours are needed.

Thus, a statement to address this suggestion in the Discussion section (see lines 481-493 in the revised manuscript).

  1. I recommend putting asterisks that represent the p value in the figures

Asterisks has been included in the figures 2 (Panels C and D), 3 (A) and 5 (C, D). A sentence in Material and Methods section has been added (line 589)

  1. Figure 5E is not described

Legend of Figure 5E has been added.

  1. Regarding figure 6I. The authors comment that they have categorized HER2-low samples that express it as low grade (1) or medium grade (2), both with low expression intensity (+), as well as being negative in FISH. Could you clarify this categorization? A 2+ sample would indicate that there is indeed expression in a significant % of cells in the sample. Is then when the result of FISH is used? Are both results taken into account? Why was a FISH performed?

Samples have been categorized as per ASCO CAP guidelines, standard in the clinic (line 337). A reference (19) has been included to help readers gain understanding of this clinical concept. As a summary:

  • IHC 3+ is considered HER2+
  • IHC 1+ or 2+ are considered HER2 equivocal and FISH is required. Positive FISH is considered HER2+. No positive FISH to HER2 samples have been included in our cohort. IHC 1+ or 2+ with negative FISH are considered HER2-low (within the HER2- category)
  • IHC 0 is considered HER2-

All these concepts are undergoing substantial review with the current development of HER2-targeting antibody-drug conjugates, but does not impact the results of this paper.

Overall, we look forward that the changes made in the revised manuscript satisfy the reviewer.

Sincerely,

Jose Angel García-Saénz and Vanesa García-Barberán